# Evaluation of Growth, Yield, and Biochemical Attributes of Bitter Gourd (*Momordica charantia* L.) Cultivars under Karaj Conditions in Iran

**DOI:** 10.3390/plants10071370

**Published:** 2021-07-05

**Authors:** Akram Valyaie, Majid Azizi, Abdolkarim Kashi, Ramaraj Sathasivam, Sang Un Park, Akifumi Sugiyama, Takashi Motobayashi, Yoshiharu Fujii

**Affiliations:** 1Department of Horticulture, Faculty of Agriculture and Natural Resources, Karaj Branch, Islamic Azad University, Karaj 3149968111, Iran; akram_vlyy@yahoo.com; 2Department of Horticulture, Faculty of Agriculture, Ferdowsi University of Mashhad, Mashhad 9177948978, Iran; 3Department of Horticulture, Faculty of Agriculture and Natural Resources Campus, University of Tehran, Karaj 7787131587, Iran; Kashi@ut.ac.ir; 4Department of Crop Science, Chungnam National University, 99 Daehak-ro, Yuseong-gu, Daejeon 34134, Korea; ramarajbiotech@gmail.com; 5Department of Smart Agriculture Systems, Chungnam National University, 99 Daehak-ro, Yuseong-gu, Daejeon 34134, Korea; 6Research Institute for Sustainable Humanosphere (RISH), Kyoto University, Kyoto 611-0011, Japan; akifumi_sugiyama@rish.kyoto-u.ac.jp; 7Department of International Environmental and Agricultural Science, Tokyo University of Agriculture and Technology, Tokyo 183-8509, Japan; takarice@c.tuat.ac.jp (T.M.); yfujii@cc.tuat.ac.jp (Y.F.)

**Keywords:** bitter melon, charantin, medicinal plant, diabetes, active ingredients, fruit yield

## Abstract

Vegetative and reproductive characteristics, fruit yield, and biochemical compounds of six bitter melon cultivars (Iranshahr, Mestisa, No. 486, Local Japanese, Isfahan, and Ilocano) were evaluated under Karaj conditions in Iran. The phytochemical properties of the cultivars were evaluated using both shade-dried and freeze-dried samples at three fruit developmental stages (unripe, semi-ripe, and ripe). There were significant differences in the vegetative and reproductive characteristics among cultivars, where cv. No. 486 was superior to most vegetative attributes. The fruit yield of cultivars varied from 2.98–5.22 kg/plant. The number of days to male and female flower appearance ranged from 19.00–25.33 and from 25–33 days, respectively. The leaf charantin content was in the range of 4.83–11.08 μg/g. Fruit charantin content varied with developmental stage, drying method, and cultivar. The highest charantin content (13.84 ± 3.55 µg/g) was observed at the semi-ripe fruit stage, and it was much higher in the freeze-dried samples than the shade-dried samples. Cultivar No. 486 had the highest (15.43 ± 2.4 µg/g) charantin content, whereas the lowest charantin content (8.51 ± 1.15 µg/g) was recorded in cultivar cv. Local Japanese. The highest total phenol content (25.17 ± 2.27 mg GAE/g) was recorded in freeze-dried samples of ripe fruits of cv. No. 486, whereas the lowest phenol content was detected in the shade-dried samples of semi-ripe fruits of Isfahan. cv. Flavonoid content was higher with the shade-drying method, irrespective of cultivar. In conclusion, considering the fruit yield and active biological compounds in the studied cultivars, cv. No. 486 should be grown commercially because of its higher yield and production of other secondary metabolites.

## 1. Introduction

Diabetes is one of the most common and dangerous chronic diseases in humans. It has been estimated that, by 2025, more than 75% of people living in developing countries will have diabetes [1]. According to global statistics in 2015, it was estimated that there were 415 million people with diabetes, and that, by 2040, one in 10 adults (642 million) will have diabetes [2]. The indicated difference reveals that the proportion of people infected with the disease in the community will be much higher than expected. The number of people with diabetes in Iran is higher than the global average [3]. For this reason, the study of new effective plant sources for curing diabetes is among the research priorities of the Ministry of Health and Medical Education of Iran.

Bitter melon, a useful medicinal plant, is used in most developing countries, particularly in Asia and Africa, as a herb, medicine, and vegetable [4]. Unripe fruit and seeds of bitter melon have been the subject of extensive research with respect to phytochemical and pharmaceutical properties [5]. The medicinal value of this plant, which can be used to treat many diseases, including diabetes, has attracted the attention of scientists worldwide [6]. All parts of this plant, including the leaves and fruits, are used in traditional medicine in Asia, Africa, and even Western countries to cure diabetes [7]. It has been reported that leaf and fruit extracts of bitter melon have antioxidant effects [8]. Fresh fruit or ground dried fruit is effective at reducing blood glucose levels [9].

Genetic and environmental factors and farming systems have significant effects on the quantity of bioactive compounds in medicinal plants. Different melon cultivars have produced types of active ingredients. In this study, bitter melon cultivars from Japan and the Philippines were evaluated to obtain information about charantin [10]. The charantin content in Japanese cultivars was higher than that in the Filipino cultivars reported by Kim et al. (2014). New plant species transferred to other areas express the same characteristics as those in the place of origin, although, by planting in a new area, changes in some characteristics, such as the duration of growth periods and phytochemicals, are expected. Studies have been carried out on the adaptability and yield of bitter melons, where it was observed that variations are more closely related to growth period duration, flowering time, fruit ripening, and other morphological and biochemical characteristics [11].

Yield is a complex attribute that is strongly influenced by environmental factors [12]. The yield potential of different varieties of a single plant species varies from one environment to another, and varieties do not show the same yield potential in different environments [11]. The content and composition of a plant’s bioactive chemicals differ from their origin to other locations because of changes in metabolic activity, which is affected by environmental factors [12]. Important environmental factors that affect the quality and quantity of bioactive compounds are light, temperature, soil, irrigation, and altitude above the free sea level [13,14]. This plant is located in India, Bangladesh, the Philippines, Thailand, Malaysia, China, Japan, eastern and southeastern Pakistan, Indonesia, Australia, the tropical regions of Africa, South America, and the Caribbean [7]. Bitter melons are grown and harvested in limited areas of Iran. However, the detection of phytochemicals from different plant parts, such as leaves and fruits, has not yet been reported. This is the first report comparing different bitter melon cultivars and their adaptability to Karaj environmental conditions in Iran, as well as the determination of effective bioactive compounds in their leaves and fruits.

## 2. Results

### 2.1. Variation of Vegetative Traits among the Cultivars

Significant differences were observed among bitter gourd cultivars for vegetative characteristics (Table 1). The differences in plant height, number of leaves per plant, leaf area, and plant dry weight were significant at the 1% level of probability, whereas the number of branches per plant was significant at the 5% level of probability.

The maximum and minimum plant heights were 355.33 and 175.33 cm for cv. Hybrid No. 486 and cv. Ilocano, respectively. This attribute plays an important role in determining the appropriate distance between rows during commercial field cultivation.

Among the cultivars, the Thai cultivar (Hybrid No. 486) produced the highest number of leaves/plant (2756), while the lowest number of leaves (1646) was observed in cv. Isfahan (Table 1). The mean leaf area of 10 leaves per plant was evaluated for each cultivar (Table 1); these were found to vary from 15.46 cm^2^ (cv. Ilocano) to 64.73 cm^2^ (cv. Iranshahr). The highest plant dry weight was observed in Hybrid No. 486 (665.66 g) and the lowest was observed in cv. Ilocano (222.33 g) (Table 1). Considering that all parts of the plant, including the leaves and fruits, are used for pharmaceuticals and traditional medicine, Hybrid No. 486 showed the best performance in terms of both leaf and fruit dry weight. The cv. Ilocano, with the lowest plant height, had the highest number of branches per plant (229.33). The lowest number of branches was observed in cv. Iranshahr (cv. palee) and cv. Local Japanese (Table 1).

### 2.2. Variation of Reproductive Traits among the Cultivars

Significant differences (*p* ≤ 0.01) were observed among the cultivars in terms of their reproductive characteristics. Male flowers appeared earlier than female flowers in all the cultivars. However, there was a 1 week difference between the cultivars in days to the appearance of the first male flower (Table 2). The first male flower appeared in cv. Local Japanese at 25.33 days after planting, which was the latest among the cultivars. The other cultivars did not show any significant differences in this regard. The first male flower appeared in Hybrid Mestisa and Hybrid No. 486 19 days after planting.

The first female flowers appeared 25 days after planting in Hybrid Mestisa and Hybrid No. 486, and 33 days after planting in cv. Isfahan. For example, the fruit yield of Hybrid No. 486 and Hybrid Mestisa, which showed shorter differences, resulted in higher fruit yield (Table 2).

#### 2.2.1. Dry Fruit Yield (g/Plant)

Analysis of variance showed that the dry fruit yield per plant and its components were significantly different (*p* ≤ 0.01) among the bitter melon cultivars. Hybrid No. 486 had the highest fruit yield per plant (0.42 kg.plant^−1^) followed by Hybrid Mestisa (0.38 kg.plant^−1^) (Figure 1). Considering the fresh and dry fruit weight per plant, Hybrid No. 486 had the highest weight, followed by Hybrid Mestisa from the Philippines (Figure 1).

#### 2.2.2. Fruit Number per Plant

The number of fruits per plant ranged from 24 to 133. The highest fruit number per plant was observed in cv. Ilocano and could be attributed to the lower ratio of male to female flowers (Table 2).

#### 2.2.3. Fruit Characteristics

There were significant (*p* ≤ 0.01) differences among bitter melon cultivars in terms of fruit length and fruit diameter at the unripe and semi-ripe stages, as well as a change in aril color at the ripe stage.

#### 2.2.4. Fruit Length

At the full ripe stage, fruit color changed from green to orange. At this stage, hybrid No. 486 had the longest fruit (32.50 cm) (Table 3).

#### 2.2.5. Fruit Diameter

The diameter of the fruit was measured at the unripe, semi-ripe, and color-changing stages of the ariels. Measurements at the completely ripe stage were not possible. The highest fruit diameter (5.19 cm) was observed in the cv. Iranshahr (cv. Palee), and the lowest (3.61 cm) belonged to cv. Ilocano.

#### 2.2.6. Fruit Density

Assessment of fruit density in different cultivars during unripe to ariel color-changing stages indicated that, during the development of the fruit, its density decreased considerably during the last stages (Figure 2). The study of the cross-sections of fruits of bitter melon cultivars in different fruit developmental stages supported this observation. At 50% of the final fruit volume, cv. Ilocano had the highest fruit density, while hybrid Mestisa had the lowest (Table 3).

#### 2.2.7. Number of Seeds per Fruit

The total number of seeds/fruit varied from 27 to 39.33 (Table 4). The highest total seeds/fruit (39.33 seeds) belonged to cv. Iranshahr (cv. palee) and the lowest (27 seeds) belonged to cv. Ilocano. There was a positive correlation between seed number per fruit and fruit size in medium-weight medicinal pumpkins. However, there was no such relationship in the heavier fruits. The redistribution of photosynthetic assimilates from fruits to seeds is a genetic attribute.

#### 2.2.8. Thousand Seed Weight

The 1000-seed weight is an indicator of seed size. The mean comparison of the 1000-seed weight of bitter melon cultivars showed that Hybrid No. 486 had the highest (216.6 g) 1000-seed weight, followed by cv. Local Japanese (214.66 g) (Table 4).

#### 2.2.9. Fruit Color

Mean comparison of color-related coefficients, including L, a, b, and ΔL, showed that there were significant differences between bitter melon cultivars for fruit color indices (Table 4). The highest L (41.36) was recorded for Hybrid Mestisa (in other words, its fruit color was brightest), while the lowest L (22.34) belonged to Hybrid No. 486 (i.e., the fruit color of this cultivar was darker than the remaining cultivars). The “L” index determines the color luminosity, where a higher “L” value is representative of the brighter color of the fruit. In our study, the lowest fruit brightness was observed in Hybrid No. 486, which was consistent with the high levels of fruit tubercle. The highest luminosity was observed in Hybrid Mestisa, which had the least tubercle on the fruit surface. Therefore, it can be expected that cultivars with fewer tubercle fruits would be brighter and have greater marketability. The mean comparison for “a” index (which indicates the color of the fruit) indicated a significant difference among the cultivars, with the largest “a” index (7.37) in Hybrid No. 486, which represents the green color of the fruit (Table 4). The “b” index determines the brightness or darkness of color; the highest “b” (21.53) belonged to Hybrid Mestisa, which indicates light green, while the lowest “b” index (11.8) was recorded for Hybrid No. 486 (Table 4). The low “b” index indicated the dark-green color of the fruit in this cultivar.

### 2.3. Phytochemical Characteristics

#### 2.3.1. Total Phenol Content

The total phenol content of fruits (Regardless of the drying method and developmental stages) was the highest (17.8 mg gallic acid equivalent (GAE) in Hybrid No. 486, followed by Hybrid Mestisa, cv. “Isfahan,” Local Japanese, and cv. Iranshahr. The Ilocano had the lowest level (10.42 mg GAE) (Table 5). The highest total phenol content in Hybrid No. 486 was obtained in freeze-dried fruits in the unripe and ripe stages (25.17 and 24.12 mg GAE, respectively). However, the lowest phenol content (3.57 mg GAE) in cv. Isfahan was in the semi-ripe stage and shade-dried samples (Table 5 and Table 6).

#### 2.3.2. Total Flavonoid Content

Investigating the amounts of flavonoids in different stages of fruit development (regardless of cultivar and drying method) showed that, with increasing developmental stages of fruit from the unripe stage, the flavonoid content decreased significantly (*p* ≤ 0.01). The highest flavonoid content in the unripe stage was 15.31 mg quercetin equivalent (mg QE) and the lowest was equal to 6.57 mg QE in the ripe fruit (Table 7). The dual interaction effects of cultivar × drying method and cultivar × development stage on fruit total flavonoid content are shown in Table 7. The results showed that the highest total flavonoid content (19.42 mg QE) was in Hybrid No. 486 in the unripe fruit stage in shade-dried samples. The lowest flavonoid content was observed in the freeze-dried cv. Local Japanese and Hybrid Mestisa in semi-ripe fruit and ripe fruit. However, the flavonoid content was lower in the ripe fruits of cv. Iranshahr (cv. Palee) for both drying methods (Table 7 and Table 8).

#### 2.3.3. Fruit Charantin Content

There was a significant difference (*p* ≤ 0.01) among the bitter melon cultivars for fruit charantin content. The dual interaction effects of cultivar × drying method and cultivar × development stage also had a significant (*p* ≤ 0.01) effect on fruit charantin content. The mean comparison of fruit charantin content is presented in Table 9. Fruit charantin content in different stages of fruit development indicated that the highest charantin was 13.84 μg/g in semi-ripe stage, while it was 12.46 μg/g in unripe fruits, and the lowest in ripe fruits. The fruit charantin content in different bitter melon cultivars subjected to different drying methods in the three fruit development stages is presented in Figure 3. The fruit charantin content in the lyophilized samples was higher than that in the shade-dried samples, irrespective of cultivar. In both methods, the highest fruit charantin content was obtained in the semi-ripe fruit (Figure 3).

The highest charantin content was detected in Hybrid no. 486 in semi-ripe fruit using the freeze-drying method. Since this cultivar had the highest fruit yield, it is suitable for cultivation under Karaj conditions. The lowest charantin content was detected in cv. Isfahan in an unripe stage in shade-dried samples and Hybrid Mestisa in the fully ripe stage using freeze-dried samples. It can be concluded that Hybrid no. 486 had high fruit yield and the highest fruit charantin content in the semi-ripe stage using freeze-dried samples; thus, it can be recommended for growth in Karaj conditions.

#### 2.3.4. Leaf Charantin Content

Mean leaf charantin content in six bitter melon cultivars is presented in Figure 4. There were significant differences between the cultivars; cv. Iranshahr with 11.08 μg/g had the highest leaf charantin content, while cv. Local Japanese had the lowest (4.83).

## 3. Discussion

The perusal of mean performance of the bitter gourd cultivars revealed that plant height, number of leaves per plant, leaf area, and plant dry weight were significant at the 1% level of probability, whereas the number of branches per plant was significant at the 5% level of probability. Significant differences in plant height of different bitter gourd cultivars reportedly previously ranged from 98 cm [15] to 575 cm [16]. It has been reported that plant leaf numbers vary with cultivars and other management practices [16,17,18,19,20]. Other researchers also showed significant differences between cultivars for leaf area, and our findings are in agreement with their results [21,22]. Moreover, the high herb-producing cultivar (Hybrid No. 486) had more green leaves and higher mean leaf area (Table 1), resulting in a higher amount of photosynthesis, thus representing a strong source for producing secondary metabolites. Differences in the number of branches among cultivars did not follow the same trend as biomass production. Islam et al. (2014), who studied 15 indigenous bitter melon landraces with small fruits, reported that the population with small and round fruits had a short main branch and numerous lateral branches compared to those plants bearing rectangular fruits [22]. This observation is in agreement with our findings, particularly for cv. Ilocano. Although the Ilocano cultivar produced the highest number of branches/plant (Table 4), plant height was the lowest in this cultivar, which negatively affected the leaf area. Iranshahr cultivars produced the lowest number of branches/plant but the larger leaf. The number of branches is the result of apical dominance, and it is important to determine the optimum plant distance on the row during cultivation. A plant’s compact canopy affects light absorption and carbon acquisition efficiency for photosynthesis and vegetative growth by changing leaf area index (LAI), which is a special genetic trait of plant cultivars, also influenced by cultural practices [20]. The presence of highly significant correlations among different plant cultivar traits indicated dependency of one character on other, and there was a negative correlation of number of branches/plant with plant height and leaf size in another report [23].

The results showed that male flowers appeared earlier than female flowers in all the cultivars. The difference in the appearance of the first female flower was much higher than that of male flowers among the studied cultivars. Therefore, cv. Local Japanese and cv. Isfahan had comparatively late flowering among the other cultivars. Fruits of the early flowering cultivars reached the harvesting stage faster. Fruit harvesting stimulated the appearance of more female flowers in all of the cultivars. The appearance of the male flower before the female flower plays a special role in pollination and fruit formation [24]. Rathod et al. (2008) reported that the number of days from direct sowing until the appearance of the first male flower varied between 37.25% and 40.23% [25]. The timing of male and female flower appearance depends on the genetic and environmental factors that are related to the adaptation of the cultivars to different regions, and some researchers have reported that this can vary from 24 to 41 days [15,21,22,26,27,28,29,30]. This is in agreement with the results of the present study, which showed 12 days difference in the appearance of male and female flowers. The time between the appearance of the first male and female flowers among the studied cultivars varied between 5.66 and 12 days after transplantation. Our observations showed that a shorter difference between the appearance of the first male and female flowers led to a higher fruit yield.

On average, the first female flower appeared at nodes 9–17 (in Philippine and Japanese cultivars, respectively) (Table 2); which agrees with Dalamu et al. (2012) who reported that the first flower appeared at the ninth node [31]. It can, thus, be concluded that the earlier appearance of female flowers at the lower nodes is strongly correlated with early ripening and fresh and dry fruit yield. Gupta et al. (2015) reported that the number of days to the appearance of the first female flower had a positive and significant correlation with some traits, including fruit yield [11]. Other researchers have also shown that male flowers appear at lower nodes compared to female flowers [14,19,20,27,31]. The number of fruits and the fruit yield/plant in bitter melon are highly dependent on the presence of male and female flowers (sex ratio). The sex ratio ranged from nine (cv. Ilocano) to 20 (cv. Local Japanese) in the different cultivars. Sex ratio strongly affected pollination and fruit formation. A greater sex ratio can provide sufficient pollen for a longer period to fertilize female flowers. Other researchers have also reported significant differences in sex ratios among different bitter melon cultivars, for example, from 6.85 to 27.29 [25,32], 7.90 to 9.77 [33], 19 to 20 [29,30], 17.99 to 33.50 [15], 10 to 30 [28], and 17.17 to 22.18 [27].

The highest fresh fruit yield (5.22 kg/plant) was obtained from Hybrid No. 486, while cv. Ilocano produced the lowest yield (2.98 kg/plant) (Figure 2). Although it produced the highest fruit number per plant (133.33), cv. Ilocano had a lower fresh fruit yield because of its small fruit size. This finding is consistent with that of Dey et al. (2006), who reported that genotypes with smaller fruits had lower fresh fruit yields [33]. The fresh fruit yield of bitter melon is economically important and depends on the variety and crop management practices. The average economic yield was 8000–10,000 kg·ha^−1^. Some cultivars may produce up to 20,000–40,000 kg·ha^−1^ [5,6,33] and even higher (25 to 80 tons) [34]. These variations can be attributed to climatic conditions, soil fertility, and the cultivar used. In Taiwan, a yield of 61–108 tons·ha^−1^ was obtained under greenhouse conditions from grafted seedlings [35], and, in India, yields from 21 to 44 tons·ha^−1^ were reported [6]. The lower yield of our research is related to the shorter growing period in Karaj, Iran, than in tropical regions such as India and the Philippines.

Our results revealed that the dry fruit yield per plant and yield attributing traits were significantly different among the bitter melon cultivars. The differences in the order of cultivars with respect to fruit yield on the basis of fresh or dry fruit yield was related to the moisture percentage of each cultivar. Several factors affect fruit water content such as fertilization [18], cultivar, fruit density, time of harvest, maturity, and texture, and it varied between 91% and 93% [36].

The current results showed that the highest fruit number per plant was observed in cv. Ilocano and could be attributed to the lower ratio of male to female flowers. No significant differences were observed among the remaining cultivars in this trait. The fruit number per plant depends on the plant’s genetics, growth conditions, and crop management practices, as reported by several studies [24,32,37,38]. Our findings are consistent with those of these researchers, except for cv. Ilocano. However, a fruit number per plant of 6.6 to 68 has been reported [3,19,26,27].

According to our results, the observed differences in fruit length can be attributed to the different stages at which the fruit length was measured. Fruit length was also reported by different researchers as 15.82–38.83 cm [27], 8.78–33 [4], 9.23–24.47 cm [15,32], and 2.91–11.60 cm [37]. Dudhat and Patel showed that nutrient management affects fruit length [38].

Our results showed that the highest fruit diameter was observed in the cv. Iranshahr (cv. Palee), and the lowest belonged to the cv. Ilocano. Other researchers reported fruit diameters of 2.2 to 7.4 cm [19,32,39], which is consistent with our findings at unripe and semi-ripe stages.

In this study, the highest and the lowest numbers of total seeds/fruit belonged to cv. Iranshahr (cv. palee) and cv. Ilocano, respectively. Moreover, there was a positive correlation between seed number per fruit and fruit size in medium-weight medicinal pumpkins. Hand pollination in *Momordica charantia* produced heavier fruit, and open pollination and fruit size were related to the number of seeds/fruit [40]. Total seeds/fruit had a wide range reported by several researchers, i.e., 7.4–28.43 [25], 7.4–28.43 [37], 15.65–33 [27], and 10.45–26.04 [22].

According to our results, the mean comparison of the 1000-seed weight of bitter melon cultivars showed that Hybrid No. 486 had the highest 1000-seed weight, followed by cv. Local Japanese. It was previously reported by many researchers that the 1000-seed weight varied among different bitter melon genotypes, that is, 58 g [40], 166 g [41], and 21.4 g [37]. Our findings are in agreement with some of these reports.

Our results of the phytochemical assay showed that the total phenol content of fruits was the highest in Hybrid No. 486, followed by Hybrid Mestisa, cv. “Isfahan,” Local Japanese, cv. Iranshahr (ee), and cv. Ilocano. A comparison of the effect of different drying methods on the phenolic content of fruit in different cultivars indicated that the freeze-drying method was superior to shade-drying. Horax et al. (2005) reported the phenolic compounds in four types of bitter melons in both oven-dried and freeze-dried samples as being between 4.55 and 82.4 mg/g dry weight [42]. They also reported the phenolic compounds in the seeds as being from 4.61 to 40.08 mg/g dry weight. They concluded that the oven-drying method was superior to freeze-drying. In another study, Horax et al. (2010) investigated the phenolic content of bitter melon pericarps and seeds. The results showed that the total phenol content in bitter melon seeds was higher than that in fruit pericarps [43]. Kubola and Siriamornpun (2008) evaluated phenolic compounds in different organs of bitter melon (leaves, stems, unripe, and ripe fruits) and showed that the leaves contained the highest total phenol content, and unripe fruit contained higher total phenol content than the ripe fruit [44]. Tan et al. (2014) grew six varieties of bitter melon in greenhouse conditions and compared the total phenol content of the fruit, showing that this trait varied from 1.5 to 7.9 mg GAE [45]. The phenolic contents measured in the present study were higher than those reported by Tan et al. (2014). The higher levels of phenolic compounds may be due to genotypic differences, climatic conditions, and crop management practices. As mentioned earlier, although secondary metabolite production is under genetic control, environmental factors also have significant effects.

According to our results, the highest total flavonoid content was obtained in Hybrid No. 486 in the unripe fruit stage in shade-dried samples. The lowest flavonoid content was observed in the freeze-dried cv. Local Japanese and Hybrid Mestisa in semi-ripe fruit and ripe fruit. However, the flavonoid content was lower in the ripe fruits of cv. Iranshahr (cv. Palee) for both drying methods. Tan et al. (2014) investigated the extraction efficiency of flavonoids from bitter melon fruits and showed that the solvent type had a significant effect on flavonoid extraction efficiency. The lowest extraction efficiency (<2 mg of retinol per gram of fruit dry weight) was obtained, while the highest flavonoid content (approximately 25 mg/g equivalent to retinol) was obtained when acetone was used. They finally suggested that the acetone solvent acted to extract the maximum flavonoids of bitter melon [45]. Therefore, the variations in the results obtained by different researchers are primarily due to the climatic conditions and secondarily due to the solvent used in the extraction of flavonoids.

The current results showed that the fruit charantin content at different stages of fruit development was highest at the semi-ripe stage, followed by unripe and ripe fruits. Secondary metabolite production is controlled by different plant pathways such as malonic acid, mevalonic acid, and shikimic acid pathways. During plant developmental stages, some genes are activated and some genes are silenced, resulting in the different secondary metabolite composition of medicinal plants [46]. Another important factor that affects secondary metabolite content and composition is the post-harvest technology, especially the drying condition and methods. During the drying period, transformation of some primary compounds to others takes place under the control of some enzymes [47]. In our research, charantin content for different drying methods indicated the superiority of lyophilized compared to shade-dried. Zhang et al. (2009) also examined the effects of the drying method on the phytochemical characteristics of bitter melons and showed that the freeze-drying method improved the phytochemical composition of the plant [48].

Kim et al. (2014) compared the charantin content of 10 Japanese and Filipino bitter melon cultivars and concluded that Japanese cultivars were superior to the Filipino cultivars. Among the cultivars studied in their research, “Peacock” (Japanese cultivar) had the highest fruit charantin content (711 μg/g), while “Trident” 357 (Filipino cultivar) had the lowest (29.8 μg/g) [48]. The findings of our study are consistent with those reported by Lee et al. (2015) and Goo et al. (2016) [49,50]. These differences in fruit charantin content in different studies are due to differences in genotypes and growing conditions [50]. The fruit total phenol, flavonoid, and charantin content, and ripe fruit color of the six bitter melon cultivars studied here indicated that cultivars with darker fruit color contained more secondary metabolites than the cultivars with brighter fruits. Habicht et al. (2011) also showed that the cultivars with white fruit had the lowest amount of saponin in comparison to the darker cultivars. Therefore, the results of this study are consistent with the findings of Habicht et al. (2011) [51].

Our results showed that cv. Iranshahr had the highest leaf charantin content, while cv. Local Japanese had the lowest. Lee et al. (2015) examined the charantin content of the Korean native population of *Momordica charantia*, “Erabu”, and Japanese Dragon cultivars and reported that the mean leaf charantin content was 144 μg/g, while the average fruit charantin was 55.27 μg/g. They stated that improved Japanese cultivars are more expensive than the native Korean population [49]. The difference between the findings of the present study and those of Lee et al. (2015) is mainly due to differences in cultivars and climatic conditions. Goo et al. (2016) showed significant differences among eight indigenous populations of bitter melons (two from Indonesia and six from India) for leaf charantin content. They reported the leaf charantin content of three Indian populations as KSI3 (5 μg/g), KSI7 (5.9 μg/g), and KSI8 (5.8 μg/g). They also confirmed that leaf charantin content was greater than that of fruits [50].

## 4. Materials and Methods

### 4.1. Plant Materials

This study was conducted in the greenhouse and research field of Seed and Plant Improvement Institute in Karaj, Iran, with geographical coordinates latitude 35°48′ N and longitude 50°58′ E, and an elevation of 1292 m above free sea level (Appendix A). Six bitter melon cultivars, namely, Hybrid No. 486 (from Thailand), Hybrid Mestisa (from the Philippines), Iranshahr (cv.Palee from India), Ilocano (from the Philippines), Isfahan, and Local Japanese, collected from different regions in Iran and other countries, were used in this study (Appendix A).

### 4.2. Plant Establishment in the Field

Seeds were sown in small pots (200 cm^3^) for seedling production. The medium was prepared using a mixture of perlite, peat moss, and soil (1:1:1). Irrigation was regularly applied to the bottom of the pots. The greenhouse temperature during the seed germination process was 20 ± 2 °C, and the relative humidity was adjusted to 55%. Seedlings were transplanted into the field at the 4–5-leaf stage. Each experimental plot consisted of one row, with a 200 cm distance between rows and 50 cm between two plants within rows. A tape irrigation system was used to irrigate the plots. Soil samples were taken from a depth of 0–30 cm before seedling planting (three samples). The results of the soil analyses are presented in Table 10. Considering the results of soil analysis (Table 10), 75 kg·ha^−1^ of urea, 150 kg·ha^−1^ of triple-super phosphate, and 100 kg·ha^−1^ of potassium sulfate were applied before planting as basal fertilizers. Humic acid (Humax) was applied 3 weeks after transplanting the seedlings in the field. Two top dressings of urea, once at the beginning of flowering (75 kg·ha^−1^) and another at the fruit setting (75 kg·ha^−1^) stages, were applied. Zinc sulfate (40 kg·ha^−1^) was applied at the beginning of the flowering stage.

### 4.3. Experimental Design and Recorded Parameters

This study was designed and implemented in two separate experiments.

Experiment 1: In the first experiment, the vegetative, reproductive, and fruit traits of bitter melon cultivars were evaluated using a randomized complete block design with three replications. Parameters at vegetative stage, i.e., plant height, number of leaves/plant, leaf area, leaf fresh weight/plant and leaf dry weight/plant, above-ground dry weight/plant, number of branches, and stem moisture content were recorded. Reproductive traits included days to the appearance of the first male and female flowers, the node number on which the first male or female flower appeared, male/female flower ratio, number of fruits/plant, total number of seeds/fruit, 1000-seed weight, fruit color-related indices (using KONICA MINOLTA’s Hunter-Leaf machine, Chiyoda City, Tokyo, Japan) at 2 cm below the fruit’s tail), fresh/unripe fruit weight, ripe fruit weight, length, diameter, weight, density, and volume of fruit were measured and evaluated. Fresh fruit yield per plant was measured using the technique described by Palada and Chang (2003) [6].

Experiment 2: In the second experiment, the effect of the time of harvesting (unripe, semi-ripe, and ripe) and drying methods (shade-dried and freeze-dried) on the phytochemical attributes of the fruit, i.e., the total phenol, flavonoids, and charantin contents were examined using factorial arrangements in a randomized block design with three replications. All means then compare statistically by Tukey’s test. Fruits of each cultivar were sampled at three different stages, unripe, semi-ripe, and ripe, following the methodology reported by Horax et al. 2010. Unripe is described as when the pericarp is still green and the seed is not fully grown yet (about 2 weeks after flowering when it is 25% of the full size) [43,52]. Semi-ripe is described as when the green pericarp and seed are fully grown (approximately 3–4 weeks after flowering when it is 50% of full size). Ripe is described as when the pericarp is yellow, and the seed arils and internal tissues are red (approximately 4–5 weeks after flowering and 100% fruit size). In addition, fruit growth parameters were evaluated when the green pericarp and seed were fully grown, the seed arils turned pink, and fruits were 80% of full size. Following sampling, the internal texture of the fruit and the development stage of the seeds were examined and recorded upon cutting the fruit (Figure 5). The freeze-drying conditions used for drying the fruit samples are listed in Table 11.

### 4.4. Phytochemical Analysis

The phytochemical characteristics of dried samples, including total phenol and flavonoid content, were measured at the Ferdowsi University of Mashhad. Freeze-drying of samples was performed in the lyophilization section of the Razi Vaccine and Serum Research Institute in Karaj. Charantin determination was performed using HPLC at Chungnam National University, Daejeon, South Korea.

### 4.5. Sample Preparation

Fruits with pulp were cut into pieces (0.5 cm) and dried in the shade at room temperature (25 ± 3 °C). Aril’s sugar percentage: After mixing the arils with a milling machine, the pulp was used for soluble solid content evaluation using a refractometer (Digital Palette PR-32 Atago, Tokyo, Japan).

### 4.6. Determination of Total Phenol Content

Extraction and determination of total phenol content were performed following the method of Horax et al. (2005) [42]. Briefly, 1 g of dried fruit was pulverized using a mortar and pestle. Then, 20 mL of methanol was added and extracted using a water bath at 65 °C for two h. The extracts were filtered using a Buchner funnel and vacuum pump, and centrifuged at 4750 rpm for 15 min. One milliliter of extract was added to 1 mL of Folin–Ciocâlteu (0.25 N), 1 mL of sodium carbonate (1 N), and 7 mL of distilled water. For the blank sample, methanol was added instead of the extract. After vortexing for 1 min, the samples were incubated for 2 h at room temperature in the dark. Finally, the absorbance of the resulting solution was measured using a spectrophotometer (Shimadzu PC-1650, Kyoto, Japan) at 726 nm.

### 4.7. Determination of Total Flavonoid Content

The total flavonoid content of the fruit was determined according to the method of Pekal and Pyrzynska (2014). One milliliter of extract was mixed with 0.5 mL of aluminum chloride, 0.5 mL of water, 0.5 mL of HCl, and 0.5 mL of sodium acetate (1 N). The samples were then incubated at room temperature for 10 min. Finally, the absorbance of the solution was measured at 425 nm using a spectrophotometer (Shimadzu PC-1650, Kyoto, Japan) [46].

### 4.8. Determination of Charantin

Extracting and measuring of charantin was done at Chungnam National University, Daejeon, South Korea following the method of Kim et al. 2014. One gram of leaf or fruit sample was mixed with 20 mL of hexane to remove the sample fats. The sample was then placed under a normal hood to evaporate the hexane. This procedure was repeated twice. Samples were then extracted with 10 mL of pure methanol using a sonicator (Branson Ultrasonic Co., Danbury, CT, USA). The extract was filtered through filter paper and dried using a vacuum evaporator (rotary evaporator) at 40 °C. Finally, the dried extract was dissolved again in 1 mL of HPLC-grade methanol. Charantin content was measured using an HPLC (NS-4000 model, Daejeon, South Korea) with a UV detector at a wavelength of 204 nm. The column used for separation was an Optimapak C-18 column (4.6 mm × 250 mm, RStech, Daejeon, South Korea) with a flow rate of 0.8 mL·min^−1^. A mobile phase of MeOH/water (98:2 *v*/*v*) was used for separation, and the injected volume of the sample was 20 μL. The identification and quantification of compounds were done by comparing peak area and the retention times with the external standard (Charantin (92.2% purity), ChromaDexInc., Irvine, CA, USA). The quantification and analysis of each sample were performed in triplicate [10].

## 5. Conclusions

According to all the results of this study, which was conducted for the first time in Iran, it can be stated that there is potential for economic production of this plant in Karaj climatic conditions, with cv. No. 486 being the best choice among evaluated cultivars in terms of vegetative traits and valuable medicinal compounds. Regarding drying of harvested fruits, although valuable medicinal compounds were better preserved in the freeze-dried samples, but due to the cost of this method and lack of access to this equipment for farmers, shade-drying is recommended. In the future, further studies are required to analyze the individual phenolic compounds and secondary metabolites found in each organ of cv. 486 by using HPLC and GC–TOFMS analysis. This study might help to provide knowledge on the utilization of strategies to increase the level of different medicinal compounds in cv. No. 486. Moreover, analyzing the gene expression of phenylpropanoid and flavonoid biosynthetic pathways will help to increase the secondary metabolite production in cv. No. 486 via the bioengineering approach.

## Figures and Tables

**Figure 1 plants-10-01370-f001:**
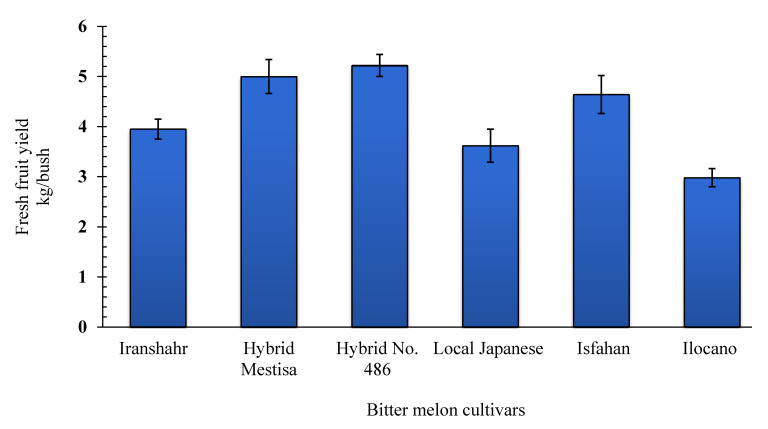
Fresh and dry fruit yield/plant (Immature stage) of six bitter melon cultivars grown in Karaj, Iran. Columns with the same letter are not significantly different at the 1% probability level using Tukey’s test.

**Figure 2 plants-10-01370-f002:**
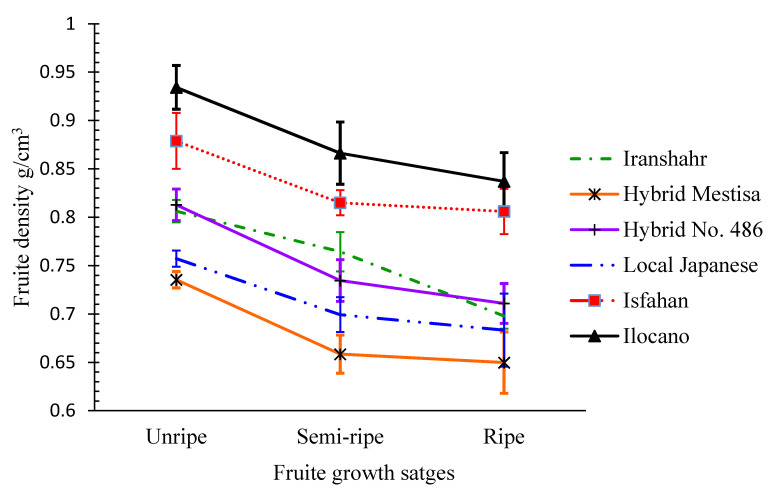
Changes in the fruit density of different bitter melon cultivars during fruit developmental stages from unripe to ariel color change stages.

**Figure 3 plants-10-01370-f003:**
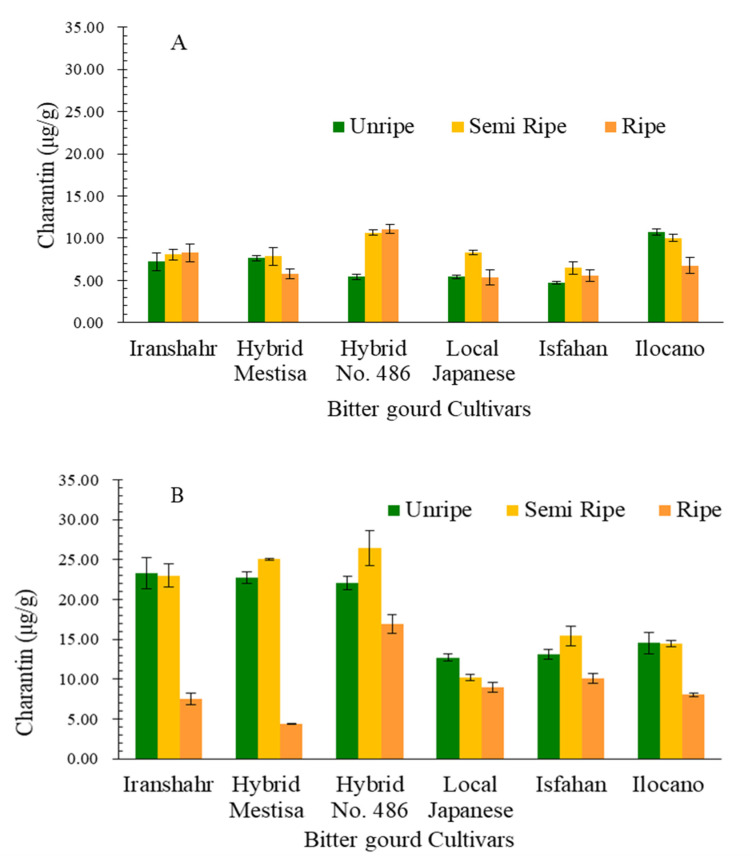
Fruits charantin content in six bitter melon cultivars during different developmental stage and drying condition (**A)**: shade-dried, (**B**): freeze-dried.

**Figure 4 plants-10-01370-f004:**
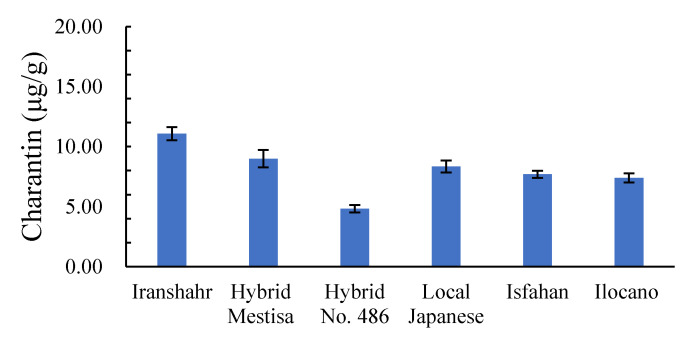
Leaf charantin content in six bitter melon cultivars grown under Karaj conditions.

**Figure 5 plants-10-01370-f005:**
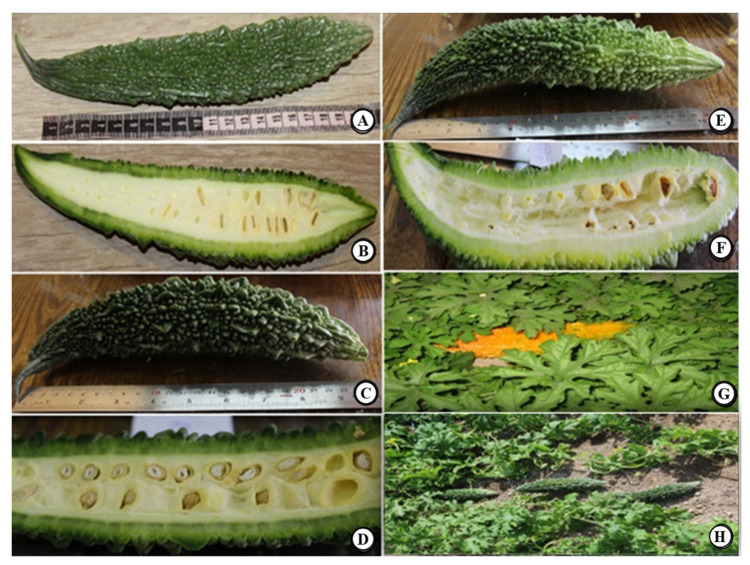
Exterior and interior fruit features at different developmental fruit stages (Hybrid No. 486 from Thailand). Unripe stage: left column (**A**,**B**); semi-ripe stage: Left column (**C**,**D**); aril color changing stage: right coumn (**E**,**F**); ripe stage: right column (**G**), growth habit (**H**).

**Table 1 plants-10-01370-t001:** Mean comparison of vegetative traits of six bitter melon cultivars.

Cultivar	Leaf No Per Plant	Leaf Area (cm^2^/10 Leaf)	Plant Dry Weight (g)	Plant Height (cm)	Number of Branches/Plant
Iranshahr	1735.33 ± 15 c	64.7 ± 2.5 a	541.00 ± 15 ab	337.00 ± 12 ab	128.00 ± 3.5 b
Hybrid Mestisa	2312.66 ± 21 ab	46.4 ± 1.6 b	433.00 ± 17 abc	311.00 ± 9 ab	170.00 ± 6.7 ab
Hybrid No. 486	2756.33 ± 25 a	60.8 ± 3.2 a	666.00 ± 23 a	355.00 ± 11 a	168.00 ± 3.5 ab
Local Japanese	2162.66 ± 14 bc	58.5 ± 2.1 a	489.00 ± 19 abc	242.00 ± 18 bc	146.00 ± 4.4 b
Isfahan	1646.66 ± 9 c	60.8 ± 2.4 a	381.00 ± 21 bc	332.00 ± 21 ab	152.00 ± 4.6 ab
Ilocano	2184.00 ± 11 ab	15.5 ± 1.6 c	222.00 ± 18 c	175.00 ± 13 c	229.00 ± 5.4 a

Means (three replications) in each column, followed by at least one letter in common are not significantly different at the 5% probability level using Tukey’s test.

**Table 2 plants-10-01370-t002:** Mean comparison of reproductive characteristics of six bitter melon cultivars.

Traits						
Cultivar	Days to First Male Flower	Days to First Female Flower	Node Number at 1st Female Flower Appear	Days Difference for Male and Female Flower Appearance	Male: Female Flower Ratio	Number of Fruits/Plant
Iranshahr	21.00 b	29.66 ab	16.00 a	8.66 ab	15.46 bc	33.18 b
Hybrid Mestisa	19.00 b	25.33 c	9.00 b	6.33 b	17.11 ab	33.07 b
Hybrid No. 486	19.33 b	25.00 c	12.6 ab	5.66 b	13.21 c	34.35 b
Local Japanese	25.33 a	31.33 ab	17.00 a	6.00 b	20.06 a	24.51 c
Isfahan	21.00 b	33.00 a	14.33 a	12.00 a	12.20 cd	32.22 b
Ilocano	19.33 b	28.33 bc	13.66 a	9.00 ab	9.03 d	133.33 a

Means (three replications) in each column, followed by at least one letter in common are not significantly different at the 5% probability level using Tukey’s test.

**Table 3 plants-10-01370-t003:** Mean comparison of the fruit characteristics of six bitter melon cultivars at different developmental stages.

	Length of Fruit (cm)				Diameter of Fruit (cm)			
	Unripe	Semi-Ripe	Aril Color Change	Ripe	Unripe	Semi-Ripe	Aril Color Change	Fruit Density (g/cm^3^)
Iranshahr	16.96 c	19.00 c	22.90 c	25.16 b	5.19 a	6.00 a	6.00 b	0.76 bc
Hybrid Mestisa	23 a	28.93 a	30.00 a	32.16 a	4.75 a	5.25 a	6.09 b	0.65 e
Hybrid No. 486	21 ab	25.00 b	30.00 a	32.50 a	5.14 a	5.77 a	6.76 a	0.73 cd
Local Japanese	18.00 bc	20.00 c	24.43 bc	27.16 b	4.81 a	5.72 a	6.33 ab	0.69 de
Isfahan	20.30 ab	23.93 b	27.26 ab	28.50 b	5.06 a	5.77 a	6.43 ab	0.81 ab
Ilocano	6.70 d	8.46 d	9.63 d	11.00 c	3.61 b	4.14 b	4.43 c	0.86 a

Means (three replications) in each column, followed by at least one letter in common are not significantly different at the 5% probability level using Tukey’s test.

**Table 4 plants-10-01370-t004:** Total seed number per fruit, 1000-seed weight, and color indices for six bitter melon cultivars.

Cultivars	Traits					
	Total Number of Seeds/Fruit	1000-Seed Weight (g)	Color Index			
			L*	a	b	∆L
Iranshahr	39.33 ± 3.1 a	179.00 ± 3.5 c	28.64 bc	−8.00 ab	13.64 b	−65.66 bc
Hybrid Mestisa	30.50 ± 2.6 abc	190.50 ± 2.7 b	41.36 a	−13.15 d	21.53 a	−52.95 a
Hybrid No. 486	27.66 ± 1.7 c	216.16 ± 1.9 a	22.34 c	−7.37 a	11.80 b	−71.96 c
Local Japanese	29.00 ± 2.4 bc	214.66 ± 4.1 a	27.21 bc	−10.85 cd	16.22 ab	−67.09 bc
Isfahan	38.00 ± 3.1 ab	157.33 ± 5.7 e	29.36 b	−8.06 ab	13.59 b	−64.95 b
Ilocano	27.00 ± 4.2 c	168.00 ± 6.2 d	32.67 b	−10.07 bc	15.63 b	−61.63 b

Means (three replications) in each column, followed by at least one letter in common are not significantly different at the 5% probability level using Tukey’s test. *CIELAB color space also referred to as L*a*b* and ∆L are color indices defined by the International Commission on Illumination (abbreviated CIE) in 1976.

**Table 5 plants-10-01370-t005:** Fruit phenol content of six cultivars of bitter melon (mg GA equivalent).

	Fruit Phenol Content (mg GA eq.)					
	Unripe		Semi-Ripe		Ripe	
Cultivars	Shade Drying	Freeze Drying	Shade Drying	Freeze Drying	Shade Drying	Freeze Drying
Iranshahr	9.56 ± 1.02 f–k	22.51 ± 1.3 a–c	5.09 ± 0.91 j–k	10.49 ± 1.05 d–k	7.96 ± 1.04 g–k	12.01 ± 1.11 c–k
Hybrid Mestisa	14.92 ± 2.01 a–j	21.30 ± 0.95 a–d	6.76 ± 0.82 i–k	7.62 ± 1.63 h–k	18.85 ± 2.13 a–g	17.34 ± 1.65 a–i
Hybrid No. 486	10.78 ± 1.7 d–k	25.17 ± 1.65 a	9.43 ± 0.56 f–k	16.56 ± 1.87 a–i	20.70 ± 2.16 a–e	24.12 ± 2.11 ab
Local Japanese	8.34 ± 0.85 g–k	17.87 ± 2.01 a–h	8.76 ± 0.47 g–k	15.47 ± 1.54 a–j	15.72 ± 1.84 a–j	13.78 ± 1.63 b–k
Isfahan	12.26 ± 1.13 c–k	20.15 ± 1.94 a–f	3.57 ± 0.21 k	13.99 ± 1.33 b–k	18.42 ± 1.38 a–h	12.22 ± 0.65 c–k
Ilocano	9.64 ± 1.1 e–k	11.54 ± 1.63 c–k	7.62 ± 0.64 h–k	6.65 ± 0.87 i–k	11.84 ± 1.12 c–k	15.21 ± 1.55 a–j

Means (three replications) in each column, followed by at least one letter in common are not significantly different at the 5% probability level using Tukey’s test.

**Table 6 plants-10-01370-t006:** Cultivar × drying method and cultivar × development stage interaction effects on fruit total phenolic content (mg GAE).

	Drying Method		Development Stage		
Cultivars	Shade Drying	Lyophilized	Unripe	Semi-Ripe	Ripe
Iranshahr	7.54 ± 1.60 d	15.00 ± 0.50 bc	16.04 ± 1.27 abc	7.79 ± 1.32 ef	9.98 ± 1.15 cdef
Hybrid Mestisa	13.51 ± 1.35 bc	15.42 ± 1.10 b	18.11 ± 1.15 ab	7.19 ± 1.28 f	18.09 ± 0.95 ab
Hybrid No. 486	13.64 ± 0.95 bc	21.95 ± 1.25 a	17.98 ± 1.50 ab	13.00 ± 0.80 bcdef	22.41 ± 1.34 a
Local Japanese	10.94 ± 1.30 bcd	15.71 ± 1.30 b	13.10 ± 1.85 bcdef	12.11 ± 0.125 bcdef	14.75 ± 1.27 bcde
Isfahan	11.42 ± 2.10 bcd	15.45 ± 0.95 b	16.21 ± 1.12 abc	8.78 ± 1.55 def	15.32 ± 1.15 bcd
Ilocano	9.70 ± 1.12 cd	11.14 ± 1.12 bcd	10.59 ± 0.85 cdef	7.13 ± 1.35 f	13.53 ± 1.40 bcdef
Mean	11.12	15.78	15.34	9.33	15.68

Means (three replications) in each column, followed by at least one letter in common are not significantly different at the 5% probability level using Tukey’s test.

**Table 7 plants-10-01370-t007:** Fruit flavonoid content of six cultivars of bitter melon (mg quercetin equivalent).

Fruit Flavonoid Content (mg Quercetin eq.)						
	Unripe		Semi-Ripe		Ripe	
Cultivars	Shade-Drying	Freeze-Drying	Shade-Drying	Freeze-Drying	Shade-Drying	Freeze-Drying
Iranshahr	15.63 ± 3.06 a–g	18.44 ± 1.85 ab	8.70 ± 0.88 d–k	5.77 ± 0.55 i–k	4.37 ± 0.35 k	4.08 ± 0.34 k
Hybrid Mestisa	18.08 ± 1.65 abc	11.05 ± 1.66 a–k	9.49 ± 0.65 d–k	3.60 ± 0.61 k	7.40 ± 0.62 g–k	3.58 ± 0.61 k
Hybrid No. 486	19.42 ± 1.71 a	16.07 ± 1.23 a–f	11.94 ± 1.04 a–k	14.07 ± 0.85 a–i	9.61 ± 0.88 c–k	9.63 ± 0.55 c–k
Local Japanese	16.93 ± 1.35 a–d	13.45 ± 1.45 a–j	7.80 ± 1.32 f–k	6.17 ± 0.55 h–k	4.28 ± 0.55 k	8.13 ± 0.81 e–k
Isfahan	16.47 ± 1.45 a–e	14.50 ± 1.66 a–h	9.11 ± 1.12 d–k	7.76 ± 0.77 f–k	5.03 ± 0.41 jk	6.46 ± 0.68 h–k
Ilocano	15.21 ± 1.66 a–g	8.51 ± 0.95 d–k	7.94 ± 0.88 f–k	11.08 ± 0.86 a–k	10.37 ± 0.87 b–k	5.86 ± 0.88 i–k

Means (three replications), in each column, followed by at least one letter in common are not significantly different at the 5% probability level using Tukey’s test.

**Table 8 plants-10-01370-t008:** Cultivar × drying method and cultivar × development stage interaction effects on fruit flavonoid content (mg QE).

Drying Method			Development Stage		
Cultivars	Shade Drying	Lyophilized	Unripe	Semi-Ripped	Ripe
Iranshahr	9.57 ± 0.85 abcd	9.43 ± 0.45 bcd	17.04 ± 1.60 ab	7.24 ± 1.12 ef	4.23 ± 0.50 f
Hybrid Mestisa	11.66 ± 0.50 abc	6.08 ± 1.20 d	14.57 ± 1.40 abc	6.55 ± 0.55 ef	5.49 ± 0.25 f
Hybrid No. 486	13.66 ± 1.10 a	13.26 ± 0.75 ab	17.75 ± 1.35 a	13.00 ± 1.50 abcd	9.62 ± 0.30 cdef
Local Japanese	9.67 ± 1.25 abcd	9.25 ± 1.10 bcd	15.19 ± 1.12 ab	6.99 ± 0.55 ef	6.21 ± 0.10 f
Isfahan	10.20 ± 1.40 abcd	9.57 ± 0.80 abcd	15.49 ± 1.50 ab	8.43 ± 0.65 def	5.75 ± 0.70 f
Ilocano	11.18 ± 1.35 abc	8.48 ± 1.15 cd	11.86 ± 0.50 bcde	9.51 ± 0.80 cdef	8.12 ± 0.35 def
Mean	10.99	9.35	15.31	8.62	6.57

Means (three replications) in each column, followed by at least one letter in common are not significantly different at the 5% probability level using Tukey’s test.

**Table 9 plants-10-01370-t009:** Cultivar × drying method and cultivar × development stage interaction effects on fruit charantin content (μg/g).

Cultivars	Drying Method		Development Stage			
	Shade-Drying	Lyophilized	Unripe	Semi-Ripe	Ripe	Mean
Iranshahr	7.87 ± 0.91 def	17.93 ± 1.37 ab	15.25 ± 1.51 bcd	15.55 ± 1.02 bc	7.91 ± 0.90 hij	12.9 ± 1.14
Hybrid Mestisa	7.09 ± 0.63 ef	17.39 ± 0.31 abc	15.21 ± 0.54 bcd	16.44 ± 0.58 b	5.06 ± 0.30 k	12.2 ± 0.47
Hybrid No. 486	9.06 ± 1.15 def	21.79 ± 1.41 a	13.72 ± 0.58 de	18.54 ± 1.36 a	14.01 ± 0.90 cde	15.4 ± 0.94
Local Japanese	6.37 ± 0.41 f	10.65 ± 0.3 def	9.07 ± 0.33 hi	9.28 ± 0.35 gh	7.18 ± 0.75 j	8.5 ± 0.47
Isfahan	5.60 ± 0.51 f	12.89 ± 0.83 bcd	8.93 ± 0.42 hij	10.97 ± 0.96 fg	7.84 ± 0.65 hij	9.2 ± 0.67
Ilocano	9.15 ± 0.56 def	12.36 ± 0.66 cde	12.61 ± 0.85 ef	12.26 ± 0.40 ef	7.38 ± 0.60 ij	10.8 ± 0.61
Mean	7.52	15.50	12.46	13.84	8.23	11.5 ± 0.71

Means (three replications) in each column, followed by at least one letter in common are not significantly different at the 5% probability level using Tukey’s test.

**Table 10 plants-10-01370-t010:** Physical and chemical properties of soil in experimental field at Karaj, Iran.

Clay	Silt	Sand	Texture	OC%	EC (Ds/m)	pH	N (total)	P (ava)	K (ava)	Fe	Zn	Cu	Mn	B
							mg/kg							
27	40	33	Clay-loam	0.63 ± 0.02	1.82 ± 0.21	7.56 ± 0.23	600 ± 15	27.6 ± 2.6	295 ± 6.3	7.97 ± 2.1	0.90 ± 0.03	2.22 ± 0.14	16.5 ± 0.3	0.44 ± 0.06

OC: Organic Carbon; EC: Electrical Conductivity, B: Boron.

**Table 11 plants-10-01370-t011:** Freeze-drying conditions.

	Factor	Level
1	Loading temperature	Room Temperature
2	Freezing rate	0.5 °C/min
3	Freezing temperature	−40 °C
4	Complete freezing duration	2 h
5	Temperature ramp rate during the primary drying	8 °C/h

## Data Availability

The data is contained within the article or Appendix A.

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
