# Peer review of "Evaluation of Growth, Yield, and Biochemical Attributes of Bitter Gourd (Momordica charantia L.) Cultivars under Karaj Conditions in Iran"

_plants, 2021, doi:10.3390/plants10071370_

Round 1
Reviewer 1 Report
Dear Authors
The manuscript is well written, has important clinical message, and should be of great interest to the readers, especially whom are interested in phytochemical extracts and treatment of diabetes using functional food and nutrients.
Overall, it is an important study, and should be considered for publication in Plants, once some issues have been resolved.
- The Abstract is comprehensive by itself. The important and essential information of the article is included.
- The overall structure of the article is well organized and well balanced. The article is written with the minimum length necessary for all relevant information.
- Appropriate and adequate references to related works are covered sufficiently in the list
Further, this is could be considered as the first report comparing different bitter melon cultivars and their adaptability to Karaj environmental conditions in Iran as well as the determination of effective bioactive 90 compounds in their leaves and fruits.
However, some issues need to be addressed :
This article does not conform to the Guide for Authors for the journal "Plants" it is submitted to. Please revise the whole manuscript in accordance with Plants journal instructions, especially the Reference in the text.
What is th common name of the plant Momordica charantia L. ? is it bitter melon or bitter apple or bitter gourd ?
Line 31: Please give the standard error for each value.
4.1. Plant materials : Insert some pictures about the your plant material : 6 melon cultivars. Insert the geographical map of which some cultivars have been harvested.
Table 9. Physical and chemical properties of soil in experimental filed at Karaj, Iran. Give the Standard Errors of each value and precise the number of repetition. Please give the reference of the Table. This comment is for all tables and figures (Bar errors need to be added).
What about the statistical analyses of your samples (determination of flavonoid, charantin, and total phenol content) ?
All the figures are not easily readable and need to be clarified (image resolution)
- Figure 3 is not clearly presented. Please could you make the comparison between different cutivars using the p value (*) instead of alphabetical letters.
- The Charantin determination was performed using HPLC. Please give more details about the conditions and parameters used.
-Lack of interpretations: Discussion that only repeats the results but does not interpret them. The researcher should have a sufficient know-how to interpret the exact reasons of the research outcome. Even if the results are out of specifications, the author should be able to critically interpret the cause in the discussion section. It is not mandatory to show positive outcomes alone. Manuscripts can support future research if they accurately interpret the root cause of the negative results.
- The conclusions ignore large portions of your findings.
Final Decision : Accept with major revision.
Looking forward to receive your revised manuscript at the earliest.
Author Response
Reviewer 1:
Q: This article does not conform to the Guide for Authors for the journal "Plants" it is submitted to. Please revise the whole manuscript in accordance with Plants journal instructions, especially the Reference in the text.
A: Thanks a lot for the comment, The manuscript ref. organized by EndNote software and I changed style on the basis of “Plant” journal instructions.
For the other parts I also checked and corrected for example Table 3 and 4 moved from M&M to results section and etc.
Q: What is the common name of the plant Momordica charantia L. ? is it bitter melon or bitter apple or bitter gourd ?
A: The common name of the plant is “bitter melon” I added in line 53.
Q: Line 31: Please give the standard error for each value.
A: The standard error of the data were added.
Q: 4.1. Plant materials: Insert some pictures about the your plant material: 6 melon cultivars. Insert the geographical map of which some cultivars have been harvested.
A: Geographical map of the investigation cite (Karaj) and other Iranian cite that the seeds were harvested was added as Figure S1 and also the colored photo of the cultivars were added as Figure S2.
Q: Table 9. Physical and chemical properties of soil in experimental filed at Karaj, Iran. Give the Standard Errors of each value and precise the number of repetition. Please give the reference of the Table. This comment is for all tables and figures (Bar errors need to be added).
A: The errors of the Tables data were added. All the tables’ data were recorded from our experiments not from other papers or books. All the tables were cited in the manuscript body text. Concerning the Bar errors in figures we used error bar in Fig 2 and 3 but in Fig1 and 4 for better difference detection we used alphabetical classification of the means on the basis of Duncuns’ multiple range test.
Q: What about the statistical analyses of your samples (determination of flavonoid, charantin, and total phenol content) ?
R: The statistical analysis and design were presented in line 500-504 as the following :
“In the second experiment, the effect of the time of harvesting (unripe, semi-ripe, and ripe) and drying methods (shade dried and freeze-dried) on the phytochemical attributes of the fruit, i.e., the total phenol, flavonoids, and charantin contents were examined using factorial arrangements in a randomized block design with three replications”
All means then compare statistically by Tukey’s test .
Q: All the figures are not easily readable and need to be clarified (image resolution)
A: The graph quality improved and also some new figure from plants were added in supplementary section.
Q: Figure 3 is not clearly presented. Please could you make the comparison between different cutivars using the p value (*) instead of alphabetical letters.
A: Fig 3 quality improved and the alphabetical letter substituted with standard errors.
Q- The Charantin determination was performed using HPLC. Please give more details about the conditions and parameters used.
A: Thank you very much the data was added to the manuscript under 4.8. section as follow:
4.8. Determination of charantin
Extracting and measuring of charantin was done at Chungnam National University, South Korea following the method of Kim et al. 2014. One gram of leaf or fruit sample was mixed with 20 mL of hexane to remove the sample fats. The sample was then placed under a normal hood to evaporate the hexane. This procedure was repeated twice. Samples were then extracted with 10 mL of pure methanol using a sonicator. The extract was filtered through filter paper and dried using a vacuum evaporator (rotary evaporator) at 40 °C. Finally, the dried extract was dissolved again in one mL of HPLC-grade methanol. Charantin content was measured using an HPLC (NS-4000 model, Korea) with a UV detector at a wavelength of 204 nm. The column used for separation is an Optimapak C-18 column (4.6 mm × 250 mm, RStech, Korea) with a flow rate of 0.8 mL min−1. A mobile phase of MeOH/water (98:2 v/v) was used for separation and the injected volume of the sample was 20 μL. The identification and quantification of compounds were done by comparing with the external standard (Charantin (92.2% purity), ChromaDexInc., Irvine, USA) peak area and the retention times. The quantification and analysis of each sample were performed in triplicate.
Q: Lack of interpretations: Discussion that only repeats the results but does not interpret them. The researcher should have a sufficient know-how to interpret the exact reasons of the research outcome. Even if the results are out of specifications, the author should be able to critically interpret the cause in the discussion section. It is not mandatory to show positive outcomes alone. Manuscripts can support future research if they accurately interpret the root cause of the negative results.
A: Thank for this important comment. I did my best for interpreting the results on the basis of my background and also previous published papers. Some results were consistent to our results and some others were in the opposite, such as results of Kim et al (2014). These differences for example in fruit charantin content of our study in comparison to other studies are due to differences in genotypes and growing conditions and discussed. Other opposite results in previous reports that I discussed in the manuscript is the results of Lee et al (2015)m concerning leaf charantin content in comparison to fruit charantin content that inserted in discussion section. The difference between the findings of the present study and those of Lee et al. (2015) is mainly due to differences in cultivar for climatic conditions. Goo et al. (2016) showed significant differences among eight indigenous populations of bitter melons (two from Indonesia and six from India) for leaf charantin content. They reported the leaf charantin content of three Indian populations; KSI3 (5 μg/g), KSI7 (5.9 μg/g), and KSI8 (5.8 μg/g). They also confirmed that leaf charantin content was greater than that of fruits [51] and this results ws in opposite of our finding.
Q: The conclusions ignore large portions of your findings.
A: The conclusion section rewrite and considered the major finding as follow:
According to all the results of this study, which was conducted for the first time in Iran, it can be stated that there is a possibility of economic production of this plant in Karaj climatic condition and cv. No. 486 should be the best choice among evaluated cultivars in terms of vegetative traits and valuable medicinal compounds. Regarding drying of harvested fruits, although valuable medicinal compounds were better preserved in the freeze dried samples, but due to its costs and lack of access to this equipment for farmers, shading drying is recommended.
Reviewer 2 Report
Review: Evaluation of growth, yield and biochemical attributes of bitter gourd (Momordica charantia L.) cultivars under Karaj conditions in Iran.
This manuscript describes the analysis of physiological parameters, such as active biochemical compounds or fruit yield in different bitter gourd cultivars under Karaj environmental conditions to make a decision on the most suitable variety for commercial purposes.
I suggest some mayor revision in these issues.
Mayor revision:
A comment for the presentation of data in general.
I see that you are analysing a huge number of variables in order to search for the best variety of the bitter gourd to be cultivated in Karaj conditions.
However, it is quite complicated to extract conclusion if the readers are asked to pay attention to all the variables that you present.
I mean that for example in figure 3, if you are focusing on searching the drying method and developmental stage that develop the highest charantin content between the varieties, could you consider representing only the variable that showed the highest values and present the statistical variances between the plant varieties?
I think you can make the effort to simplify data in the same way for phenol and flavonoid content.
Finally, you can show the results for the rest of the conditions in supplementary data.
- Paragraph between line 268 and 275:
In which table did you represent the data “Hybrid Mestisa had high fruit yield”? In figure 1, the highest dry fruit yield was for Hybrid No. 486. In addition, the highest charantin content in fruit was showed for Hybrid No. 486.
Please, revise the conclusions in this paragraph.
- Figure 3. The resolution of this figure, in general, is very bad.
In Iranshahr variety, the statistical significance analysis is missing.
Table 5: please explain between which variables you calculated the statistical variance.
- Why figure 4 and figure 5 are placed after the material and methods section whereas all the rest of the figures are placed in the results section?
Minor revision:
Figure 1: figure 1 is too small.
Table 1: Could you adapt the writing in the first line of the table in two rows?
Table 2: It misses to highlight in bold the last two traits in the table.
Table 5: “Ripe”, in the first second line of the table is out of place.
Table 7: “Drying method”, in the first line of the table is out of place.
Table 8: it misses a comma in 13.72.
Thank you
Author Response
Reviewer 2
This manuscript describes the analysis of physiological parameters, such as active biochemical compounds or fruit yield in different bitter gourd cultivars under Karaj environmental conditions to make a decision on the most suitable variety for commercial purposes.
I suggest some mayor revision in these issues.
Mayor revision:
A comment for the presentation of data in general.
I see that you are analysing a huge number of variables in order to search for the best variety of the bitter gourd to be cultivated in Karaj conditions.
However, it is quite complicated to extract conclusion if the readers are asked to pay attention to all the variables that you present.
I mean that for example in figure 3, if you are focusing on searching the drying method and developmental stage that develop the highest charantin content between the varieties, could you consider representing only the variable that showed the highest values and present the statistical variances between the plant varieties?
I think you can make the effort to simplify data in the same way for phenol and flavonoid content.
Finally, you can show the results for the rest of the conditions in supplementary data.
- Paragraph between line 268 and 275:
In which table did you represent the data “Hybrid Mestisa had high fruit yield”? In figure 1, the highest dry fruit yield was for Hybrid No. 486. In addition, the highest charantin content in fruit was showed for Hybrid No. 486.
Please, revise the conclusions in this paragraph.
A: Thank you for this precise review. You are right It was corrected as: It can be concluded that Hybrid no. 486 had high fruit yield and also the highest fruit charantin content in the semi-ripe stage using freeze-dried samples and could be reccommend to be grown in Karaj conditions.
- Figure 3. The resolution of this figure, in general, is very bad.
In Iranshahr variety, the statistical significance analysis is missing.
Table 5: please explain between which variables you calculated the statistical variance.
Q:- Why figure 4 and figure 5 are placed after the material and methods section whereas all the rest of the figures are placed in the results section?
A: All these figs transferred to the results section.
Minor revision:
Figure 1: figure 1 is too small.
Q: Table 1: Could you adapt the writing in the first line of the table in two rows?
A: I adapted the first line in two rows.
Table 2: It misses to highlight in bold the last two traits in the table.
A: Thanks for your comments. I bolded the last two traits.
Table 5: “Ripe”, in the first second line of the table is out of place.
A: Thanks for your comments. I adjusted (centered) the “Ripe” header.
Table 7: “Drying method”, in the first line of the table is out of place.
A: I redesigned the table.
Table 8: it misses a comma in 13.72.
A: Thanks for the comments. I corrected the data.
Round 2
Reviewer 1 Report
The manuscript is well written, has important scientific message, and should be of interest to the readers. Overall, it should be considered for publication in PLANTS, once the statistical issue has been resolved.
- The figures are not complete (error bar not presented) or are not clear enough to read (Figure 1, 2 and 4).
- Figure 2 : need to change the unit of the Y axis in order to make the curve more clear. You need to change also the Y axis unit of the Figure 4.
- Table 5 Phenol and fruit flavonoid content of six cultivars of bitter melon. Please insert the SD of each value and the number of repetition.
- Please add a a section of statistical analysis in the Methods.
- 4.8. Determination of charantin Extracting and measuring of charantin was done at Chungnam National University, South Korea following the method of Kim et al. 2014. Please insert the number of reference.
- 5. Conclusions : conclusions must clearly present the new insights or understanding that have been gained from the work, and any implications for future research. Please add a new paragraph related to future perspectives of your study.
- it is suitable to add a list of abreviation before the introduction section.
Author Response
Review 1 Second Round
The manuscript is well written, has an important scientific message, and should be of interest to the readers. Overall, it should be considered for publication in PLANTS, once the statistical issue has been resolved.
- Q: The figures are not complete (error bar not presented) or are not clear enough to read (Figure 1, 2, and 4).
A: Fig 1 was reorganized and error bars were added
- Q: Figure 2: need to change the unit of the Y-axis in order to make the curve more clear. You need to change also the Y-axis unit of Figure 4.
A: Y-Axis in Fig 2 was changed for the curve to be more clear
A: Y-Axis in Fig 4 was changed for the curve to be more clear
- Q: Table 5 Phenol and fruit flavonoid content of six cultivars of bitter melon. Please insert the SD of each value and the number of repetitions.
- A: I added the SD of all data in Table 5 and because the data were more than the table capacity I divided this table into two separate tables 5 and 6 for Fruit phenol and flavonoid content respectively.
- All data are means of three data repetition >I added this repletion under each table.
- Q: Please add a section of statistical analysis in the Methods.
- A: The section was added to the manuscript under 4-9 as:
4.9. Statistical analysis
- The first experiment was conducted using a randomized complete block design with three replications and the vegetative, reproductive, and fruit traits of six bitter melon cultivars were evaluated. In the second experiment, the effects of harvest time (unripe, semi-ripe, and ripe) and drying methods (shade dried and freeze-dried) on the phytochemical attributes of the fruit, were examined using a factorial arrangement on the basis of the randomized block design with three replications. The data were subjected to ANOVA by using Jump 8 software. The means comparison was done by Tukey’s test and the significant differences were presented. Data means ± SD were presented in the tables and figures.
- Q:8. Determination of charantin Extracting and measuring of charantin was done at Chungnam National University, South Korea following the method of Kim et al. 2014. Please insert the number of references
- A: The number of references for this part was added.
- Q: 5. Conclusions: conclusions must clearly present the new insights or understanding that have been gained from the work, and any implications for future research. Please add a new paragraph related to future perspectives of your study.
A: The conclusion revised accordingly and a new paragraph added for future research as the following:
- A: According to all the results of this study, which was conducted for the first time in Iran, it can be stated that there is a possibility of economic production of this plant in Karaj climatic condition and cv. No. 486 should be the best choice among evaluated cultivars in terms of vegetative traits and valuable medicinal compounds. Regarding drying of harvested fruits, although valuable medicinal compounds were better pre-served in the freeze-dried samples, due to its costs and lack of access to this equipment for farmers, shading drying is recommended. In the future, further studies are required to analyze the individual phenolic compounds and secondary metabolites found in each organ of cv. 486 by using HPLC and GC-TOFMS analysis. This study might help to provide knowledge on the utilization of strategies to increase the level of different medicinal compounds in cv No. 486. Moreover, analyzing the gene expression of phenylpropanoid and flavonoid biosynthetic pathways will help to increase the secondary metabolites production in cv No. 486 by the bioengineering approach.
- it is suitable to add a list of abbreviations before the introduction section.
There are not many abbreviates in the manuscript. Would you please determine them clearly?
Reviewer 2 Report
Dear Author,
In the first round of revision I suggested a mayor revision for the presentation of data in general.
"I see that you are analysing a huge number of variables in order to search for the best variety of the bitter gourd to be cultivated in Karaj conditions. It looks very brave from you.
However, it is quite complicated to extract conclusion if the readers are asked to pay attention to all the variables that you present.
I mean that for example in figure 3, if you are focusing on searching the drying method and developmental stage that develop the highest charantin content between plant varieties, could you consider representing only the drying methodology and developmental stage that showed the highest values and represent the statistical variances between the plant varieties?
I think you can make the effort to simplify data in the same way for phenol and flavonoid content.
Finally, you can show the results for the rest of the conditions in supplementary data"
I did not find any response from you.
Author Response
Review 2 Second Round
Dear Author,
In the first round of revisions, I suggested a major revision for the presentation of data in general.
"I see that you are analyzing a huge number of variables in order to search for the best variety of the bitter gourd to be cultivated in Karaj conditions. It looks very brave of you.
However, it is quite complicated to extract conclusion if the readers are asked to pay attention to all the variables that you present.
I mean that for example in figure 3, if you are focusing on searching the drying method and developmental stage that develop the highest character content between plant varieties, could you consider representing only the drying methodology and developmental stage that showed the highest values and represent the statistical variances between the plant varieties?
I think you can make the effort to simplify data in the same way for phenol and flavonoid content.
A: Thanks a lot for this good recommendation. Fig 3 was reorganized and changed to two separate graphs that will be very more informative than the previous format. For phenol and flavonoids, the table was divided into two separate tables for simplification of understanding.
Finally, you can show the results for the rest of the conditions in supplementary data"
I did not find any response from you.
Dear Reviewer I revised the manuscript in the first round as the following:
Reviewer 2 First Round
This manuscript describes the analysis of physiological parameters, such as active biochemical compounds or fruit yield in different bitter gourd cultivars under Karaj environmental conditions to make a decision on the most suitable variety for commercial purposes.
I suggest some mayor revision in these issues.
Mayor revision:
A comment for the presentation of data in general.
I see that you are analysing a huge number of variables in order to search for the best variety of the bitter gourd to be cultivated in Karaj conditions.
However, it is quite complicated to extract conclusion if the readers are asked to pay attention to all the variables that you present.
I mean that for example in figure 3, if you are focusing on searching the drying method and developmental stage that develop the highest charantin content between the varieties, could you consider representing only the variable that showed the highest values and presents the statistical variances between the plant varieties?
I think you can make the effort to simplify data in the same way for phenol and flavonoid content.
A: the Charantin content that is important of the plant phytochemical was reorganized in fig 3.
Finally, you can show the results for the rest of the conditions in supplementary data.
- Paragraph between line 268 and 275:
Q: In which table did you represent the data “Hybrid Mestisa had high fruit yield”? In figure 1, the highest dry fruit yield was for Hybrid No. 486. In addition, the highest charantin content in fruit was showed for Hybrid No. 486.
Please, revise the conclusions in this paragraph.
A: Thank you for this precise review. You are right It was corrected as: It can be concluded that Hybrid no. 486 had high fruit yield and also the highest fruit charantin content in the semi-ripe stage using freeze-dried samples and could be recommended to be grown in Karaj conditions.
- Figure 3. The resolution of this figure, in general, is very bad.
The Fig 3 was completely reorganized
Q: In Iranshahr variety, the statistical significance analysis is missing.
A: It was corrected
Table 5: please explain between which variables you calculated the statistical variance.
Q:- Why figure 4 and figure 5 are placed after the material and methods section whereas all the rest of the figures are placed in the results section?
A: All these figs transferred to the results section.
Minor revision:
Figure 1: figure 1 is too small.
A: It was revised
Q: Table 1: Could you adapt the writing in the first line of the table in two rows?
A: I adapted the first line in two rows.
Table 2: It misses to highlight in bold the last two traits in the table.
A: Thanks for your comments. I bolded the last two traits.
Table 5: “Ripe”, in the first second line of the table is out of place.
A: Thanks for your comments. I adjusted (centered) the “Ripe” header.
Table 7: “Drying method”, in the first line of the table is out of place.
A: I redesigned the table.
Table 8: it misses a comma in 13.72.
A: Thanks for the comments. I corrected the data.
Thanks a lot for all your time that you spent for reading and adding your valauable comments that improved my manuscript.
I did my best for revising the manuscript on the basis of your and other reviewer comments and if there are and comments for improving the manuscript plane let me know again.
Any help would be much appreciated
With the kindest regards